# Sub-Clinical Effects of Outdoor Smoke in Affected Communities

**DOI:** 10.3390/ijerph18031131

**Published:** 2021-01-28

**Authors:** Thomas O’Dwyer, Michael J. Abramson, Lahn Straney, Farhad Salimi, Fay Johnston, Amanda J. Wheeler, David O’Keeffe, Anjali Haikerwal, Fabienne Reisen, Ingrid Hopper, Martine Dennekamp

**Affiliations:** 1School of Public Health and Preventive Medicine, Monash University, Melbourne 3004, Australia; t_odwyer@hotmail.com (T.O.); lahn.straney@moa.com.au (L.S.); farhad.salimi@monash.edu (F.S.); dokeeffe@livingpositivevictoria.org.au (D.O.); anjali_haik@hotmail.com (A.H.); ingrid.hopper@monash.edu (I.H.); Martine.Dennekamp@epa.vic.gov.au (M.D.); 2Environmental Health, Menzies Institute for Medical Research, University of Tasmania, Hobart 7000, Australia; fay.johnston@utas.edu.au (F.J.); Amanda.Wheeler@acu.edu.au (A.J.W.); 3Behaviour, Environment and Cognition Program, Mary MacKillop Institute for Health Research, Australian Catholic University, Melbourne 3000, Australia; 4Climate Science Centre, CSIRO Oceans and Atmosphere, Aspendale 3195, Australia; Fabienne.Reisen@csiro.au; 5Environmental Public Health Unit, Environment Protection Authority Victoria, Melbourne 3053, Australia

**Keywords:** smoke, PM_2.5_, landscape fire, bushfire, biomarkers, FeNO, neutrophils, white cell count

## Abstract

Many Australians are intermittently exposed to landscape fire smoke from wildfires or planned (prescribed) burns. This study aimed to investigate effects of outdoor smoke from planned burns, wildfires and a coal mine fire by assessing biomarkers of inflammation in an exposed and predominantly older population. Participants were recruited from three communities in south-eastern Australia. Concentrations of fine particulate matter (PM_2.5_) were continuously measured within these communities, with participants performing a range of health measures during and without a smoke event. Changes in biomarkers were examined in response to PM_2.5_ concentrations from outdoor smoke. Increased levels of FeNO (fractional exhaled nitric oxide) (β = 0.500 [95%CI 0.192 to 0.808] *p* < 0.001) at a 4 h lag were associated with a 10 µg/m^3^ increase in PM_2.5_ levels from outdoor smoke, with effects also shown for wildfire smoke at 4, 12, 24 and 48-h lag periods and coal mine fire smoke at a 4 h lag. Total white cell (β = −0.088 [−0.171 to −0.006] *p* = 0.036) and neutrophil counts (β = −0.077 [−0.144 to −0.010] *p* = 0.024) declined in response to a 10 µg/m^3^ increase in PM_2.5_. However, exposure to outdoor smoke resulting from wildfires, planned burns and a coal mine fire was not found to affect other blood biomarkers.

## 1. Introduction

With climate change, wildfires in Australia are forecast to increase in frequency and severity [1,2,3]. Wildfires present physical risks to individuals and communities from both direct exposure to intense heat and flames, but also from widespread exposure to smoke [4]. Planned (prescribed) burns are conducted to reduce fire fuel loads in bushland and areas surrounding communities [5]. These are designed to reduce the risk of catastrophic wildfires [6,7]. Both wildfires and planned burns are a common occurrence in Australia, and exposure to smoke from wildfires or planned burns is inevitable for the majority of Australians [8].

Wild fire and planned burn smoke contains a variety of inorganic and organic compounds, as well as airborne particulates [9]. Particulate matter with a median aerodynamic diameter smaller than 2.5 micrometres (PM_2.5_) is a significant component of the smoke [10]. The small size of these fine particles allows them to penetrate deeply into the lungs [11]. These fine particulates have been linked with chronic health conditions, such as asthma, chronic obstructive pulmonary disease (COPD), ischaemic heart disease (IHD) and lower respiratory infections (LRI) [12].

While the acute effects of urban background PM_2.5_ are well documented for premature mortality, hospital admissions, emergency presentations and ambulance call-outs [12], the acute effects of PM_2.5_ on biomarkers of systemic and airway inflammation are less well understood, particularly in the context of smoke from wild fires. Exposure to smoke from wild fire smoke has been associated with an increased risk of out of hospital cardiac arrests and IHD [13]. Most studies involving the health effects of wild fires focus on discrete outcomes and events such as mortality and hospital episodes [14,15,16]. However, there has been limited evidence of the acute effects of exposure to outdoor smoke on populations. This study investigated the effect of short-term exposures to wild fire, planned burn, and coal mine fire smoke on markers of inflammation.

## 2. Materials and Methods

We performed a short-term panel study of rural Victorian residents during the prescribed burns season and winter period over four consecutive years (Table 1). The detailed methodology has been published previously [17]. We present here a brief summary of the methods.

### 2.1. Participants

Communities likely to be impacted by smoke from planned burns were identified in conjunction with the Victorian Department of Environment, Land, Water and Planning (DELWP), the agency responsible for conducting planned burns. Three towns were identified as suitable study locations in Victoria: Warburton, Traralgon and Maffra/Heyfield [17] (Figure 1). Residents aged 18 and over were recruited through random digit dialing to identify interested individuals, with follow-up telephone calls, community advertising and letter box drops of study information. The aim was for half the sample to be over 65 years of age, as older age is a risk factor for adverse health outcomes from air pollution. There were no exclusion criteria based on current health or medical conditions.

### 2.2. Study Period

The study was conducted from Autumn 2013 to Autumn 2016.

### 2.3. Health Assessments

Participants attended two appointments for clinical testing, one during a period with no known source of outdoor smoke (the clean air assessment) and one during a smoke event. Clinical tests included:Blood tests, for markers of inflammation and coagulation, specifically high sensitivity C-reactive protein (CRP), fibrinogen, and a full blood examination (including hemoglobin, total and differential white cell and platelet counts) [17].Airway inflammation test, measuring fractional exhaled nitric oxide (FeNO) using a NiOx unit (Aerocrine AB, Solna, Sweden) [18].

### 2.4. Exposure Assessment

At a central location in each of the three study areas, an E-sampler aerosol monitor (Met One Instruments Inc., Grants Pass, OR, USA) was set up. The E-sampler measured continuous PM_2.5_ concentrations by light scattering, and collected gravimetric measurements on filters, which were used to determine a calibration factor.

### 2.5. Statistical Analysis

Comparisons between men and women were made with Pearson χ^2^ or Fisher’s exact tests. Generalized estimating equations (GEE) were fitted to assess the relationships between the biomarkers and PM_2.5_ [19,20]. The GEE calculated the average change in clinical measures for each 10 μg/m^3^ increase in PM_2.5_ with 95% confidence intervals (this was a realistic increase). Data were analyzed using Stata statistical software (Version 12.1, StataCorp, College Station, TX, USA).

Markers of lung and systemic inflammation (FeNO, total white cell (WCC), neutrophil, basophil counts, and high sensitivity C-reactive protein (CRP)) were the continuous outcomes (dependent variables). The exposure variables were hourly averaged PM_2.5_ concentrations. Analyses were performed for the following lag periods: 4 h (PM_2.5_ concentration in the 4 h prior to the test), 8 h, 12 h, 24 h and 48 h.

The parameter estimates from the GEE models may be interpreted as proportional changes in the levels of individual biomarkers. We calculated the changes in biomarker levels associated with exposure per 10 µg/m^3^ PM_2.5_ from outdoor smoke.

All analyses controlled for known confounders of temperature and humidity. Secondary analyses also controlled for individual smoking status, age and history of asthma. All analyses were conducted using Stata (version 14.1; StataCorp, College Station, TX, USA). *p*-values < 0.05 were considered statistically significant [21].

### 2.6. Ethics Approval

This study was approved by the Monash University Human Research Ethics Committee CF12/3097-2012001570 and the Human Research Ethics Committee of the University of Tasmania, reference number H0013022. All participants provided written, informed consent.

## 3. Results

There was a total of 207 participants enrolled in the full study with a subset of 183 who completed repeat measurements included in these analyses (Appendix A). The participants’ mean age (SD, min and max) was 63.5 (12.2, 26 and 92) years and 60% were female (Table 2). Just under half (46%) of the participants were aged over 65 years. A similar proportion had ever smoked tobacco for at least one year. Inhabitants had lived in the study region for a median of 22 (IQR 10 to 38 years) and the majority (69%) were from Warburton. Common co-morbidities included asthma, COPD, ischemic heart disease, heart failure and diabetes. The men were significantly older than the women, less likely to be current smokers and more likely to have heart failure or other heart conditions.

There was significant variation in the exposure to PM_2.5_ from the different smoke sources. Planned burns and wildfire smoke were only recorded in Warburton (Figure 2 and Figure 3) due to limited burning seasons across the 4-year study period. No smoke from any source was detected in Maffra/Heyfield during the study period, as scheduled planned burns did not proceed during this time due to local weather conditions. In 2014, there was a coal mine fire near Traralgon, and the air quality in Traralgon was impacted by the smoke from these fires during our study (Figure 4).

The fractional exhaled nitric oxide (FeNO) showed positive relationships with increasing concentrations of PM_2.5_ from outdoor smoke (resulting from wild fires, planned burns or a coal mine fire) for the preceding 4-, 12-, 24- or 48-h (Table 3). Total white cell (WCC) counts showed significant declines at 24-h lag period and neutrophils at 24- and 48-h lag periods associated with PM_2.5_ exposures from outdoor smoke events (Table 4). No significant changes associated with any exposure were seen for eosinophils, monocytes and platelets, fibrinogen or C-reactive protein (data not shown). The stratified analysis for the different sources of the smoke (coal mine fire, planned burn and forest fire smoke) showed positive associations between FeNO and exposure to coal mine fire smoke for the 4-h lag period (Table 3).

## 4. Discussion

This panel study showed that outdoor smoke (smoke from planned burns, wildfires and a coal mine fire) had an impact on biomarkers, including total white cell and neutrophil counts, as well as increasing the fractional exhaled nitric oxide consistent with systemic and airway inflammation.

The results for FeNO were consistent with previous studies, indicating that PM_2.5_ was associated with eosinophilic airway inflammation [22,23,24,25]. Our results confirmed these findings, showing positive associations between PM_2.5_ from outdoor smoke and FeNO.

There was less consistency between the results for the blood markers in our study and other studies. Previous studies found increases in CRP [26] and fibrinogen associated with PM_2.5_ from urban background air pollution, including traffic and woodsmoke [27,28,29,30,31]. It has also been shown in response to transient exposure to PM_2.5_ from biomass smoke exposure [32,33] in Finland and North America. This may indicate different components of these particulate fractions from alternative sources may produce different biological responses, differences in exposure profiles, or it may indicate that smoke from wildfires produces different changes in biomarkers depending on the type of vegetation that burns. Equally, different responses of biomarkers to PM_2.5_ exposure could be related to duration of exposure or lack of statistical power.

Mean levels of urban background PM_2.5_ observed by Huttenen et al. [32] were lower (8.7 µg/m^3^) than in our study, however blood samples were taken bi-weekly, which may have increased sensitivity to changes in biomarkers. Adetona et al. observed PM_2.5_ TWA levels of 338 µg/m^3^ and 240 µg/m^3^ in their exposed sample of firefighters, with samples being taken immediately following shifts [33]. There were cross-shift increases in IL8, CRP, serum amyloid A and segmented neutrophils in peripheral blood.

As Huttunen et al. [32] conducted their study in patients with heart disease, the effect of biomass smoke may have been more pronounced than in our sample. However, our panel was composed mainly of elderly individuals, with a high prevalence of heart disease. Equally, Adetona et al. [33] may have detected an increase in CRP due to the fire fighters’ close proximity to the smoke source and the very high levels of exposure. However, the strength of this interaction could also have been mitigated by a healthy worker effect. Finally, increased CRP may be a sign of other inflammatory responses not related to smoke, perhaps masking any true effects.

Previously, total WCC has been found to be associated with urban background PM_10_ [34]. This may be due to different cellular responses to larger particulate fractions, or from differences in particle composition. Ghio et al. [35] demonstrated increased levels of neutrophils in bronchial and bronchoalveolar lavage sampled in participants exposed to PM_2.5_ from wood smoke. Decreases in neutrophils and total WCC count may indicate cells moving out of the peripheral blood and into the lungs in response to inflammation. This could be investigated further through bronchoalveolar lavage or analysis of exhaled breath for neutrophil markers.

These results highlight the potential sub-clinical changes resulting from outdoor smoke exposures. Previous work relating to landscape fires has mainly measured clinical outcomes such as cardiac arrests and asthma attacks. Our findings suggest that individuals with asthma should try to reduce their exposure to smoke during wildfires or planned burns [36]. Importantly, this study also provides a crucial insight into the sub-clinical changes which may occur in the lead up to an individual suffering an overt adverse health effect. Further research is needed to determine if this finding is repeated under similar conditions.

### Strengths and Limitations

This study used the paired data from individuals to provide an objective measurement of the impact of PM_2.5_ on selected biomarkers. By measuring biomarkers, we were able to demonstrate objective changes in inflammation and clinically relevant endpoints. As these measurements were sub-clinical, our study provided an important indication of subtle health changes during smoke-related PM_2.5_ exposure.

The main limitation of this study was the difficulty in obtaining a consistent exposure to smoke due to meteorological variation. Changes in local weather conditions often resulted in planned burns not occurring, or the smoke not impacting the town. This may have reduced the chance of finding a potential association between smoke from planned burns and changes in biomarkers. On the days when planned burns were occurring nearby, there were frequently negligible amounts of smoke present in the study area. This was likely due to burn protocols designed to reduce the impact of smoke on communities as much as is practicable. Equally, the changes only demonstrated correlation and not necessarily causation. As the panel study relied on repeated measures of returning participants, there may be a healthy volunteer bias. Another limitation was the time between paired measurements. Due to changes in the planned burn schedule, there were gaps between the initial ‘clean’ measurement and the “smoke” measurement of several weeks or months. Although we have demonstrated similar responses to different sources of smoke, we accept that the findings should be generalized to other settings with caution.

However, this study has identified health impacts of planned burn smoke on FeNO and white cell counts. Further research should focus on the potential impact of smoke-related PM_2.5_ on vulnerable populations, particularly individuals with pre-existing cardiovascular and respiratory conditions. In order to determine if the reduction in WCC and neutrophils numbers is due to these cells migrating from the peripheral blood to the lungs, future studies are recommended of these effects on more highly exposed populations, such as firefighters working on wild fires or planned burns.

## 5. Conclusions

Outdoor smoke from wildfires, planned burns and a coal mine fire were associated with increased levels of FeNO, but decreased neutrophils and total WCC in peripheral blood. This suggests that PM_2.5_ may cause increased airway inflammation. This may have significant clinical implications for individuals with pre-existing respiratory conditions or compromised immune systems. However, there was no evidence of systemic inflammation.

## Figures and Tables

**Figure 1 ijerph-18-01131-f001:**
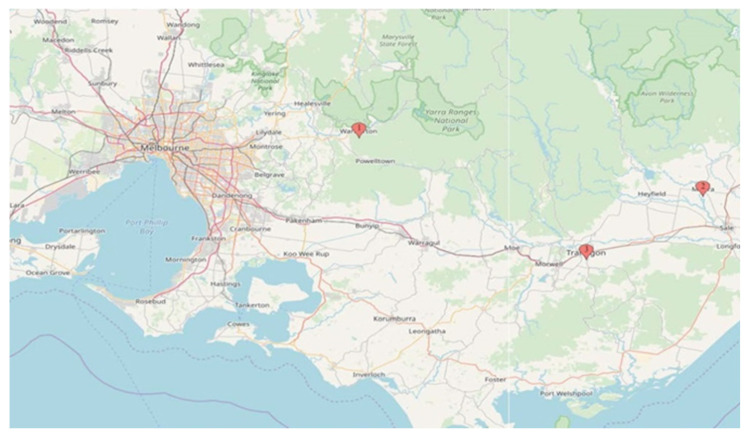
Study locations: Victoria, Australia.

**Figure 2 ijerph-18-01131-f002:**
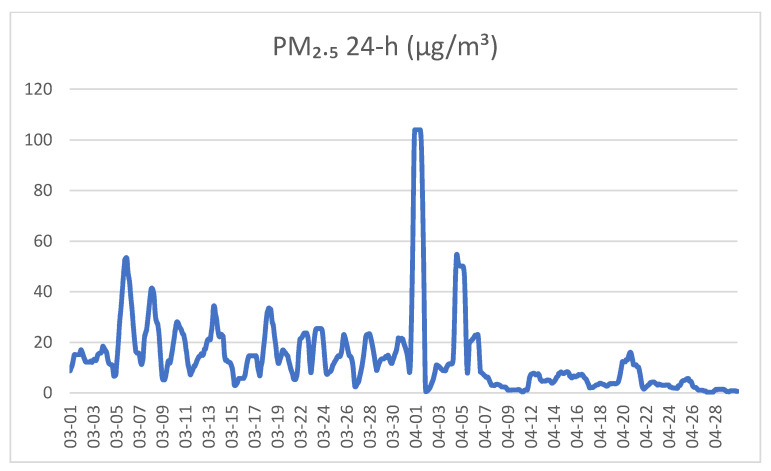
Average 24 h concentrations of PM_2.5_ (µg/m^3^) measured in Warburton between 1 March and 30 April 2016. Health assessments were conducted on 1 April 2016 and 19–21 April 2016. Note that planned burns were conducted near Warburton during the study.

**Figure 3 ijerph-18-01131-f003:**
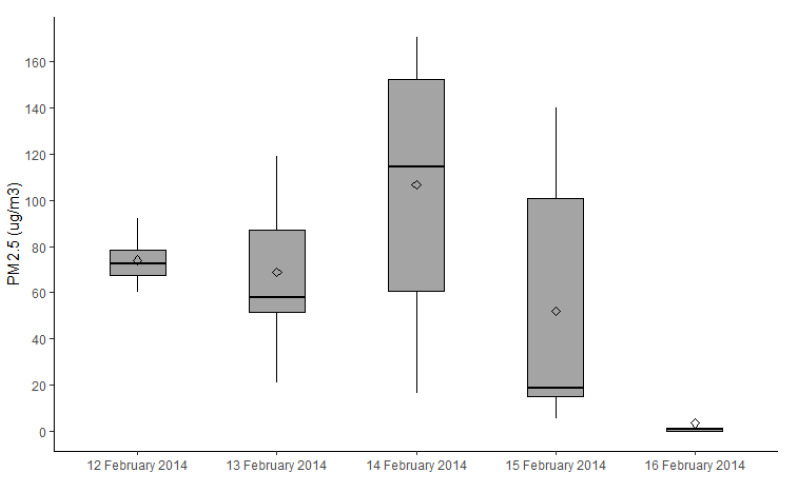
Boxplots showing median, quartiles and extreme concentrations of PM_2.5_ (µg/m^3^) as measured on the day’s health assessments were conducted in Warburton in 2014. Note that wildfires only occurred near Warburton during this study period.

**Figure 4 ijerph-18-01131-f004:**
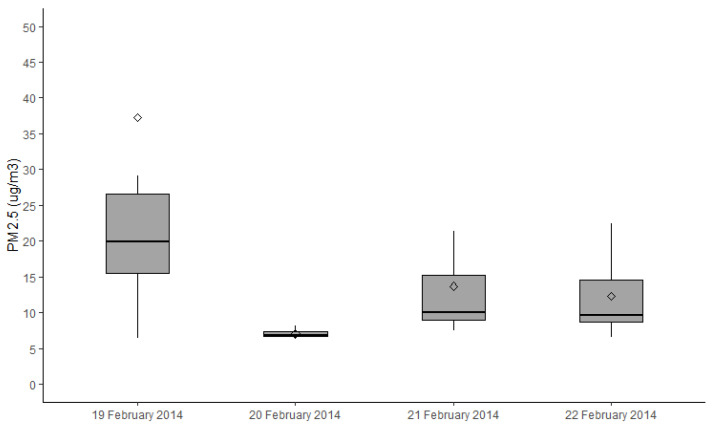
Boxplots showing median, quartiles and extreme concentrations of PM_2.5_ (µg/m^3^) as measured on the day’s health assessments were conducted in Traralgon. Note that the coal mine fire only impacted Traralgon during the study.

**Table 1 ijerph-18-01131-t001:** Number of assessments conducted per year and exposure type. Note that no clean air assessments were conducted in 2016. Assessments were carried out on those participants who were tested in Warburton in 2015 when there was no smoke.

Location and Year	Number of Assessments	Type of Smoke	Number of Assessments	Type of Smoke
Warburton 2013	14	Planned burn	10	No smoke
Warburton 2014	44	Wild fire	39	No smoke
Traralgon 2014	42	Coal Mine Fire	29	No smoke
Maffra 2014	21	No smoke	14	No smoke
Warburton 2015	7	Planned burn	78	No smoke
Warburton 2016	55	Planned burn		
Total	183		170	

**Table 2 ijerph-18-01131-t002:** Demographics and clinical characteristics of study participants.

Characteristic	Women Total (*n* = 110)	Proportions	Men Total (*n* = 73)	Proportions	*p*-Value
Age > 65 years	40	36.0%	44	60.3%	0.001
Current regular smoker	15	13.6%	3	4.1%	0.034
Smoked at all last month	33	30.0%	20	27.4%	0.75
Ever smoked	41	37.2%	32	43.8%	0.72
Asthma diagnosed	17	15.5%	12	16.4%	0.86
Asthma attack in the last 12 months	6	5.5%	0	0%	0.028 *
Asthma medication	10	9.1%	8	11.0%	0.97
COPD	4	3.6%	4	5.6%	0.80
Other respiratory condition	6	5.5%	5	6.8%	0.94
Hypertension	41	37.3%	28	38.4%	0.88
Angina	4	3.6%	7	9.6%	0.18
High Cholesterol	31	28.2%	21	28.8%	0.93
Myocardial infarction or coronary event	5	4.5%		9.6%	0.23 *
Heart Failure	1	0.9%	5	6.8%	0.039 *
Arrhythmia	14	12.7%	10	13.7%	0.85
Stroke or TIA	2	1.8%	5	6.8%	0.12 *
Other heart condition	3	2.7%	8	11.0%	0.028 *
Diabetes	14	12.7%	9	12.3%	0.94
Self-reported cold or flu symptoms	18	16.5%	10	13.7%	0.61
Immune modulators	3	2.7%	0	0.0%	-
Non-Steroidal anti-inflammatories	2	1.8%	1	1.4%	-
Anti-platelet medication	8	7.2%	8	10.9%	-
Asthma Inhalers/preventers	10	9.1%	8	10.9%	-
Antihistamines	2	1.8%	0	0.0%	-

* Fisher’s exact test.

**Table 3 ijerph-18-01131-t003:** Changes in fractional exhaled nitric oxide (FeNO) (ppb) Regression coefficients (β) and 95% confidence intervals per 10 µg/m^3^ PM_2.5_ adjusted for temperature, humidity, smoking status, asthma diagnosis and age.

Exposure Type	Lag Period (h)	*β*	95% CI	*p*
All	4	0.500	(0.192 to 0.808)	<0.001
Planned Burn Smoke	4	0.335	(−0.012 to 0.681)	0.058
Wildfire Smoke	4	0.644	(0.447 to 0.842)	<0.001
Coal Mine Fire	4	1.533	(0.461 to 2.605)	0.005
All	12	0.308	(0.028 to 0.588)	0.031
Planned Burns Smoke	12	0.269	(−0.014 to 0.553)	0.063
Wildfire Smoke	12	1.027	(0.816 to 1.239)	<0.001
Coal Mine Fire Smoke	12	0.196	(−1.467 to 1.859)	0.817
All	24	0.381	(−0.036 to 0.798)	0.073
Planned Burns Smoke	24	0.497	(−0.050 to 1.044)	0.075
Wildfire Smoke	24	1.073	(0.846 to 1.299)	<0.001
Coal Mine Fire Smoke	24	0.538	(−1.431 to 2.506)	0.593
All	48	0.344	(−0.154 to 0.842)	0.176
Planned Burns Smoke	48	0.761	(−0.165 to 1.686)	0.107
Wildfire Smoke	48	0.789	(0.539 to 1.040)	<0.001
Coal Mine Fire Smoke	48	−0.148	(−2.150 to 1.854)	0.885

**Table 4 ijerph-18-01131-t004:** Changes in total white cell count (WCC) and neutrophil counts Regression coefficients (β) and 95% confidence intervals per 10 µg/m^3^ PM_2.5_ adjusted for temperature, humidity, smoking status, asthma diagnosis and age.

Exposure Type	Outcome	Exposure Period (h)	β	95% CI	*p*
All	WCC	24	−0.088	(−0.171 to −0.006)	0.036
Planned Burns	WCC	24	−0.069	(−0.178 to 0.041)	0.218
Wildfire Smoke	WCC	24	−0.108	(−0.235 to 0.018)	0.094
Coal Mine Fire	WCC	24	−0.203	(−0.605 to 0.200)	0.324
All	WCC	48	−0.092	(−0.192 to 0.007)	0.070
All	Neutrophils	24	−0.077	(−0.144 to −0.010)	0.024
Planned Burns	Neutrophils	24	−0.065	(−0.156 to 0.026)	0.162
Wildfire Smoke	Neutrophils	24	−0.083	(−0.185 to 0.019)	0.112
Coal Mine Fire	Neutrophils	24	−0.147	(−0.469 to 0.175)	0.370
All	Neutrophils	48	−0.081	(−0.162 to −0.001)	0.048

## Data Availability

The data presented in this study are available on reasonable request from the corresponding author. The data are not publicly available due to ethical requirements.

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
