# Peer review of "Sub-Clinical Effects of Outdoor Smoke in Affected Communities"

_ijerph, 2021, doi:10.3390/ijerph18031131_

Round 1

Reviewer 1 Report

Dear authors

My main concerns are related to not having inclusion and exclusion criteria for the sample enrolled.

Second, by having only two classes of ages below and above 65 years old, there might have a huge gap in ages. We only have access to mean age that was of 63.5, would be important to have minimum and maximum.

Also, it would be important to have on the text the total number of participants enrolled.

Figure 3 and 4 should have the same scale once they are measuring the same thing.

Would prefer to observe results of cell count and neutrophil in figures instead of a table, easier to read.

All et al. in the manuscript must be in italic and with a dot at the end. 

Would review the conclusions once when there is inflammation there is an increase in white cell count.

Yours faithfully

Author Response

Reviewer 1

My main concerns are related to not having inclusion and exclusion criteria for the sample enrolled.

Details of the inclusion and exclusion criteria were included in the section ‘Participants’ [lines 66 – 70].

Second, by having only two classes of ages below and above 65 years old, there might have a huge gap in ages. We only have access to mean age that was of 63.5, would be important to have minimum and maximum.

Line 113 includes the mean and SD of the participants’ ages. We have now included the minimum and maximum ages as well.

Also, it would be important to have on the text the total number of participants enrolled.

The details of the total number of participants has now been added to line 112.

Figure 3 and 4 should have the same scale once they are measuring the same thing.

We appreciate that the scale for the PM2.5 would benefit from being on the same scale to compare the locations.  However, the great difference in the two concentrations means that it would be very difficult to interpret Figure 4 details. As such, we have retained the current figures.

Would prefer to observe results of cell count and neutrophil in figures instead of a table, easier to read.

We attempted to present the results of the white cell count models in a figure, but found that this was actually more difficult to read than Table 4.  We would prefer to retain the current Table.

All et al. in the manuscript must be in italic and with a dot at the end. 

These formatting corrections have been made.

Would review the conclusions once when there is inflammation there is an increase in white cell count.

We are not sure what the reviewer is suggesting.

Reviewer 2 Report

The experimental work is based on a previously established methodology. However, it seems pertinent to specify the methodology adopted in practical terms, as the one mentioned in table 1 is not enough to clarify the reader about the methodological assumptions adopted in the investigation. The weather conditions associated with the measurement period are not mentioned, as this is an extremely important reality for the results obtained. It is suggested that the information collected be specified, namely on the wind direction verified at the measurement sites.

The results presented, although clear, are displayed consecutively in figures, without making any intermediate analysis. As a point to improve, the finding that the presentation of the results related to the assessment of the smoke concentration needs a major review and improvement. It will also be necessary to deepen the presentation of the results of the Health assessment, and these are duly grounded in the discussion.

Conclusions are clear, although succinct. The article needs significant improvements in order to improve its scientificity.

Author Response

Reviewer 2

The experimental work is based on a previously established methodology. However, it seems pertinent to specify the methodology adopted in practical terms, as the one mentioned in table 1 is not enough to clarify the reader about the methodological assumptions adopted in the investigation. The weather conditions associated with the measurement period are not mentioned, as this is an extremely important reality for the results obtained. It is suggested that the information collected be specified, namely on the wind direction verified at the measurement sites.

We agree that local weather conditions were important.  Unfortunately, the wind direction details at each site were not measured and as such cannot be included in Table 1.

The results presented, although clear, are displayed consecutively in figures, without making any intermediate analysis. As a point to improve, the finding that the presentation of the results related to the assessment of the smoke concentration needs a major review and improvement. It will also be necessary to deepen the presentation of the results of the Health assessment, and these are duly grounded in the discussion.

We are not sure what, if any additional analysis is being requested by the reviewer.

Conclusions are clear, although succinct. The article needs significant improvements in order to improve its scientificity.

The authors are unclear exactly what additional details are being requested by the reviewer. This has not been a concern of the other reviewers and as such we would prefer to retain our presentation of results.

Reviewer 3 Report

Sub-clinical effects of outdoor smoke in affected communities.

The objective of this study was to examine effects of outdoor smoke from planned burns, wildfires and coal mine fires by assessing biomarkers in an exposed and predominantly older population. In general the concept of the manuscript is interesting and the Readers can benefit from the results presented in the paper. The only one limitation is that data are only important for the study area and the translation of the results to different population can be difficult.  

Some remarks:

Abstract:

The information about what kind of biomarkers were used should be mentioned. As later the Authors’ present the results of changes in some of the biomarkers. In my opinion the whole range of biomarkers examined should be mentioned.

Methods:

As there were no exclusion criteria on current health or medical conditions how we can link the health parameters (inflammatory biomarkers) to current exposure. Maybe some of the inhabitants are living in this areas for a short period of time so how the exposure can be associated with the examined outcome?

The information about the data collected in the questionnaire should be added.

Results:

It will be more informative if the p-value will be added to the Table 2.

Additionally the information why those biomarkers were used should be added.

Author Response

Reviewer 3

The objective of this study was to examine effects of outdoor smoke from planned burns, wildfires and coal mine fires by assessing biomarkers in an exposed and predominantly older population. In general the concept of the manuscript is interesting and the Readers can benefit from the results presented in the paper. The only one limitation is that data are only important for the study area and the translation of the results to different population can be difficult.  

We would challenge the reviewer’s assumption that the results are not transferrable to other locations. We have demonstrated that a range of smoke sources have similar health outcomes and for the health outcomes reported in the literature we have similar findings.  However we have added this as a potential limitation [lines 223-225].

Some remarks:

Abstract:

The information about what kind of biomarkers were used should be mentioned. As later the Authors’ present the results of changes in some of the biomarkers. In my opinion the whole range of biomarkers examined should be mentioned.

We have included a brief description of the biomarkers as suggested [line 18].

Methods:

As there were no exclusion criteria on current health or medical conditions how we can link the health parameters (inflammatory biomarkers) to current exposure. Maybe some of the inhabitants are living in this areas for a short period of time so how the exposure can be associated with the examined outcome?

Participants with current health or medical conditions were included to make the results more generalisable to the general population.  The statistical models controlled for a history of asthma.  Inhabitants had lived in the study region for a median of 22 years (IQR 10 to 38 years).  This information has been added to the results [lines 115-116].

The information about the data collected in the questionnaire should be added.

Details of the baseline questionnaire have now been included in the Supplementary Information.

Results:

It will be more informative if the p-value will be added to the Table 2.

Table 2 presented descriptive statistics of the participants. As requested, p-values have now been included and a comment added to the results about differences between men and women [lines 116-120].  The numbers on specific classes of medication were too small to permit further analysis.

Additionally the information why those biomarkers were used should be added.

Some rationale for the inclusion of biomarkers of inflammation is now presented in the introduction [line 48].  Additional details were included in the already published protocol [17].

Round 2

Reviewer 2 Report

Although the authors did not answer or perceive all the questions asked, with the changes made, the article meets the minimum requirements for publication